# Myocarditis: Etiology, Pathogenesis, and Their Implications in Clinical Practice

**DOI:** 10.3390/biology12060874

**Published:** 2023-06-17

**Authors:** Emil Brociek, Agata Tymińska, Andrea Silvio Giordani, Alida Linda Patrizia Caforio, Romuald Wojnicz, Marcin Grabowski, Krzysztof Ozierański

**Affiliations:** 1First Department of Cardiology, Medical University of Warsaw, 02-097 Warsaw, Poland; emil.brociek@gmail.com (E.B.); marcin.grabowski@wum.edu.pl (M.G.); krzysztof.ozieranski@wum.edu.pl (K.O.); 2Cardiology, Department of Cardiac Thoracic Vascular Sciences and Public Health, University of Padova, 35-100 Padova, Italy; andreasilvio.giordani@aopd.veneto.it (A.S.G.); alida.caforio@unipd.it (A.L.P.C.); 3Department of Histology and Cell Pathology in Zabrze, School of Medicine with the Division of Dentistry, Medical University of Silesia, 40-055 Katowice, Poland; rwojnicz@sum.edu.pl

**Keywords:** myocarditis, inflammatory cardiomyopathy, heart failure, dilated cardiomyopathy, immunosuppressive therapy

## Abstract

**Simple Summary:**

Myocarditis is an inflammatory disease of the myocardium which can lead to serious short-term and long-term sequalae such as sudden cardiac death or dilated cardiomyopathy. It is caused by infectious and non-infectious factors; however, our understanding of processes that govern its pathogenesis is limited. The aim of this review is to summarize available evidence regarding the etiology and pathogenesis of myocarditis, as well as to outline the impact that they have on disease course and patient management.

**Abstract:**

Myocarditis is an inflammatory disease of the myocardium caused by infectious or non-infectious agents. It can lead to serious short-term and long-term sequalae, such as sudden cardiac death or dilated cardiomyopathy. Due to its heterogenous clinical presentation and disease course, challenging diagnosis and limited evidence for prognostic stratification, myocarditis poses a great challenge to clinicians. As it stands, the pathogenesis and etiology of myocarditis is only partially understood. Moreover, the impact of certain clinical features on risk assessment, patient outcomes and treatment options is not entirely clear. Such data, however, are essential in order to personalize patient care and implement novel therapeutic strategies. In this review, we discuss the possible etiologies of myocarditis, outline the key processes governing its pathogenesis and summarize best available evidence regarding patient outcomes and state-of-the-art therapeutic approaches.

## 1. Introduction

Myocarditis is defined as an inflammatory disease of the myocardium diagnosed by established histological, immunological and immunohistochemical criteria [1]. The etiopathogenesis of myocarditis is complex as it is caused by a variety of infectious (i.e., viruses, bacteria, fungi, parasites) and non-infectious (i.e., organ-specific or systemic immune-mediated disease, drugs, vaccines, toxins) factors [2]. Currently, viral etiology is considered to be the most common in western countries, even though the pathogenic role of certain types of viruses has been questioned [3,4]. Myocarditis is thought to occur more frequently in males and at a young age [5,6]. Its manifestations vary significantly, ranging from mild symptoms to fulminant presentation with cardiogenic shock, acute heart failure (HF) and life-threatening arrhythmias as well as chronic HF and dilated cardiomyopathy (DCM) phenotype [7,8]. The clinical course of the disease is also heterogeneous, since some patients recover quickly, while in others, the acute phase of the disease is followed by chronic inflammation of the myocardium possibly leading to DCM [2,9]. Furthermore, particularly in patients presenting with infarct-like symptoms and preserved ventricular function, myocarditis often resolves spontaneously without specific treatment and no persisting myocardial injury, whereas in patients presenting with recognized markers of worse prognosis (e.g., reduced left-ventricular ejection fraction [LVEF], severe arrhythmias, hemodynamic instability), outcomes such as cardiac death, unremitting HF and heart transplantation are more common [6,10,11,12]. Years of combined clinical observations and experimental research have led to the elaboration of a triphasic model of myocarditis (Figure 1) [9,13,14,15,16,17,18,19].

Management of myocarditis poses a major challenge to clinicians due to the heterogeneity in clinical presentation, infrequent use of endomyocardial biopsy (EMB), which is the diagnostic gold standard, and limited treatment options supported by suboptimal evidence [2]. The understanding of myocarditis has greatly improved in recent years, but many questions still remain to be answered. As it stands, it is unclear why some patients progress to DCM, while others spontaneously recover. The therapeutic significance and prognostic utility of etiopathogenesis, patients’ genetic background and mechanisms participating in the immune response are still not entirely understood. Such data, however, are essential for accurate prognosis, risk assessment and personalized care. Moreover, a deep understanding of mechanisms governing the onset and progression of the disease is necessary for the development of novel therapeutic strategies. In this review, we summarize available evidence regarding the etiopathogenesis of myocarditis, as well as discuss how it impacts patient management and outcomes.

## 2. Clinical Presentation, Diagnostic Approach and Assessment of Disease Etiology

### 2.1. Clinical Presentation

Myocarditis has a heterogenous clinical presentation, as it can range from mild symptoms (e.g., chest pain, palpitations) to life-threatening acute HF, cardiogenic shock and ventricular arrhythmias [2,5,8]. It occurs most frequently in young males; however, females more often have a complicated clinical presentation [6]. Based on registry data, chest pain appears to be the most common patient complaint, with dyspnea being the second [6,20,21]. Fever is also present in over half of patients. In the largest registry to date, 26.6% of patients had presentation complicated by left ventricular systolic dysfunction, ventricular arrhythmias or cardiogenic shock, and 57.5% had ST-segment elevations on electrocardiogram (ECG), which is considered to be the most common ECG abnormality among patients with myocarditis [6].Therefore, it should be noted that patients with myocarditis can mimic acute myocardial infarction or acute pericarditis at presentation. In up to 80.5% of cases, patients present with prodromal symptoms, frequently suggestive of respiratory or gastrointestinal tract infections [6]. Patients with complicated presentations (e.g., decreased LVEF, ventricular arrhythmias) have a less favorable prognosis and are at a higher risk of cardiac events.

### 2.2. Utility of Diagnostic Tests

Current HF guidelines of the European Society of Cardiology (ESC) state that a 12-lead ECG, laboratory tests (e.g., troponins, natriuretic peptides, full blood count), echocardiography and cardiac magnetic resonance (CMR) are mandatory in all patients with suspected myocarditis [22]. ECG and laboratory tests are frequently abnormal in myocarditis, but the abnormalities are neither specific nor sensitive, and their absence cannot rule out the disease [2,5,6,20,21,22,23]. Echocardiography is also neither highly specific nor sensitive; however, it plays a significant role in the differential diagnosis of the patient (e.g., to rule out structural heart disease). Similarly, invasive coronary angiography or computed tomography angiography may be used to rule out coronary artery disease.

Out of non-invasive diagnostic tests, CMR is the preferred one, as it possesses the ability to assess inflammation and cardiac fibrosis, making it highly sensitive when using the 2018 modified Lake Louise Criteria [4,24,25]. However, its sensitivity varies depending on the extent of cell necrosis and clinical presentation, making it not entirely reliable in every clinical situation [26]. Moreover, CMR does not possess the ability to confirm or rule out the presence of infectious agents in the myocardium and cannot characterize immune cell infiltrates, which are the key to patient’s prognosis and management. Therefore, EMB still remains the diagnostic gold standard as it allows clinicians to establish a definitive myocarditis diagnosis in every clinical presentation, enables the assessment of the presence of infectious factors in the myocardium (most importantly viral genomes) and permits the characterization of immune cell infiltrates [2,5]. According to the 2013 Position Statement of the ESC, EMB should be considered in every patient with suspected myocarditis, and while CMR can be performed prior to EMB in clinically stable patients, it does not replace EMB and should never delay it, especially in life-threatening presentations [2].

## 3. Viral and Virus-Induced Immune-Mediated Myocarditis

### 3.1. Overview of Viruses Associated with Myocarditis

Viruses are widely considered to be a key factor in the pathogenesis of myocarditis due to the widespread presence of viral genomes in EMB samples collected from patients with myocarditis [27]. Studies report presence of viral genomes in the myocardium of almost 70% of patients with idiopathic DCM, and in nearly 30% of such patients, multiple viral agents were present [28]. Viral myocarditis is diagnosed in the presence of histological evidence of myocarditis and concurrent presence of viral genome in cardiac tissue samples, confirmed by positive polymerase chain reaction (PCR) [2]. However, available evidence suggests that viruses can also induce myocarditis in the absence of direct cardiotoxicity, doing so instead via virus-mediated inflammatory response during infection (autoimmune myocarditis) with no viral genome present in EMB [4].

Despite the fact viruses are the most studied etiological agents in myocarditis, clear evidence on the exact pathological mechanisms is lacking. This is due in large part to the variety of infectious agents that contribute to the disease and their heterogenous characteristics (e.g., different cell tropism), resulting in many putative pathological processes [29]. Available evidence suggests that Parvovirus B19 (B19V), which has an endothelial cell tropism, is the most frequently identified species in active myocarditis or DCM on EMB [28,30,31,32]. Other commonly found viruses include the following: Enteroviruses and Adenoviruses, cardiotropic; human herpesvirus type 6 (HHV-6), Epstein–Barr Virus (EBV) and Cytomegalovirus (CMV), lymphotropic; Hepatitis C virus (HCV), human immunodeficiency virus (HIV) and Influenza viruses, cardiotoxic [4,28,33]. More recently, it has been proposed that coronaviruses, in particular, severe acute respiratory syndrome coronavirus 2 (SARS-CoV-2), may play a role in the development of myocarditis. However, potential mechanisms responsible for cardiac injury in patients hospitalized with coronavirus disease 2019 (COVID-19) remain elusive [34,35]. Viruses frequently associated with myocarditis along with their key characteristics are presented in Table 1 [4,28,33].

### 3.2. Virus-Mediated Myocardial Injury

Because of the complex nature and variability between different etiopathogeneses of viral myocarditis, they need to be discussed in multiple steps. While phase 2 and 3 of the triphasic model (Figure 1) remain challenging to elucidate, phase 1 (i.e., infection of the cardiac tissue and direct virus-mediated damage) has been better investigated using animal models and human data. The process causing direct virus-induced myocardial injury appears to mostly consist of cardiomyocyte infection, subsequent replication in the infected cells and ultimately cell death. Such behavior is exhibited by adenoviruses and coxsackieviruses, which have been associated with myocarditis for decades, and both utilize the coxsackievirus-adenovirus receptor (CAR) to infect cardiomyocytes [32,36,37]. The relevance of CAR in myocarditis is highlighted by the fact that CAR-deficient mice are protected from Coxsackievirus B3-induced myocarditis and pancreatitis [38]. A large portion of experimental data regarding the pathogenesis of viral myocarditis are derived from Coxsackievirus B3 (CVB3)-infected mouse models, which have demonstrated the ability of CVB3 to induce cytopathic effects potentially leading to necrosis or apoptosis of infected cells [39,40,41]. Some of the proposed mechanisms leading to cardiomyopathy include the activity of viral protease 2A, which has been shown to disrupt the sarcolemma of cardiomyocytes and induce cardiomyopathy [42,43]. Moreover, CVB3-infected *Rag-1^−/−^* mice, which completely lack T and B cells, experienced uncontrolled viral replication in the heart and severe tissue damage, suggesting an important role of direct cardiomyocyte damage induced by viruses [44,45]. Importantly, not all mouse strains develop myocarditis after inoculation with CVB3, indicating the important role of host complex polygenic background in the pathogenesis of the disease [23].

On the other hand, B19V is a member of the erythroparvovirus genus that replicates in erythroid progenitor cells [46,47]. However, the infection of endothelial cells has also been demonstrated, and available evidence suggests that endothelial cell dysfunction plays a central role in B19V-induced myocarditis [30,46,48,49,50]. Coronary arterioles, venules and capillaries are lined with endothelium, and therefore, their infection is possible, given the concurrent presence of globoside (blood group antigen P) which acts as a B19V receptor, with required co-receptors in the form of α5 β1-integrin and Ku80 on the cell surface [51,52,53,54]. After cell entry, the mechanisms that lead to endothelial dysfunction appear intricate. In vitro studies have shown that the non-structural protein (NS1) increases the expression of pro-inflammatory cytokines, is potentially cytotoxic and induces apoptosis [55,56,57,58]. Furthermore, viral capsid protein (VP1) is thought to modulate the immune response after cell infection, induce endothelial dysfunction, as well as possibly facilitate cell proliferation [53,59,60]. VP1 may also be responsible for the facilitation of cell entry by binding to a coreceptor [59]. Persistent inflammation of the endothelium and subsequent endothelial dysfunction lead to the impairment of microcirculation and subsequent cardiomyocyte necrosis [29]. Crucially, B19V is also found in non-inflamed hearts, and therefore, its role as a bystander and possible etiological agent causing myocarditis remains uncertain and is yet to be fully understood [30]. Some authors hypothesize that coinfection with other viruses may be necessary for the initiation of B19V replication [61,62]. It is also important to distinguish between latent infection without replicative activity and actively replicating viruses. Transcriptional activity of B19V has been proposed to possibly determine the pathogenic character of B19V in myocarditis as it has been found to be crucial for altered gene expression in cardiomyopathy hearts and to possibly influence patient outcomes [63,64,65]. Quantitative determination of B19V load on EMB has also been proposed to be crucial for determining a direct role of the virus in the maintenance of myocardial inflammation. In 2010, Bock et al. suggested a viral load of more than 500 genome equivalents (ge) per microgram in EMB as a clinically relevant threshold [30].

The exact mechanisms behind myocardial injury induced by other viruses are not well understood. The presence of herpesviruses is not uncommon in patients with myocarditis or DCM [27,28,33]. This is not surprising, considering their ability to cause latent infections and high prevalence among adults [66,67]. Currently, limited data are available from mouse models infected with murine gamma herpesvirus-68 (MHV-68) and murine cytomegalovirus (MCMV). Interestingly, BALB/c mice infected with MHV-68 showed signs of myocardial necrosis, while C57BL/6 mice, as well as B- and T-cell-deficient B6-(Rag1)™ mice, did not, despite the presence of very high viral loads in B6-(Rag1)™ [68,69]. Moreover, mice studies have shown that MCMV can infect myocytes in vitro and impair their function; however, in vivo, virus titers in mice hearts were low and not significantly correlated with the severity of myocarditis [70]. These findings support the hypothesis that it is not the direct virus-induced lysis of cardiomyocytes that causes cardiomyocyte necrosis and the development of chronic myocardial inflammation, but rather the subsequent immune response to the infection and patient’s genetic profile. Furthermore, mice studies reaffirm the possibility of MCMV causing latent infection of the heart, which can lead to false associations between the presence of a viral genome and myocarditis [70]. Data from EMB sample analysis suggest that HHV-6A may infect cardiomyocytes, while HHV-6B is only found in endothelial cells [71,72]. Chromosomally integrated HHV-6 was shown to be able to replicate inside cardiomyocytes of infected individuals [72].

SARS-CoV-2 has been recognized as a cause of myocardial injury; however, due to very limited histological data, the exact processes involved in this phenomenon remain unknown [73,74,75]. It is important to note that myocarditis is a different entity to myocardial injury, and since both may appear during a viral infection, differential diagnosis should be made carefully following international guidelines [76]. Moreover, cardiomyocyte damage indicated by isolated elevated troponin levels does not have to be associated with ongoing myocardial inflammation [35]. The recently published COVID-HEART study has shown that in patients with COVID-19 and myocardial injury, myocarditis pattern on cardiac magnetic resonance (CMR), as defined by the Lake Louise criteria, is in fact far less common than infarction and microinfarction patterns [77]. This suggests that microangiopathic and macroangiopathic thrombosis is probably the more common cause of myocardial injury in COVID-19 patients. Hypothesized causes of myocardial injury also include direct cardiomyocyte infection through angiotensin-converting enzyme 2 (ACE2) receptors causing subsequent cell death, as well as excessive, maladaptive immune response [35]. Currently, it is unclear whether direct cardiomyocyte lysis or degeneration are responsible for cardiomyocyte damage, as there have been reports of SARS-CoV-2 cardiomyocyte infections with lymphocytic and macrophage infiltrates present, but no cardiac necrosis [78,79,80]. There is evidence from autopsies and EMBs performed on patients with a reported diagnosis of myocarditis which revealed the presence of SARS-CoV-2 in cardiomyocytes, necrosis as well as pronounced macrophage and minimal T cell infiltrates [81]. Notably, such infiltrate, however, is not typical for myocarditis, and different analyses support the notion that interstitial macrophage infiltrates appear to be much more common in COVID-19 [75]. It is not clear whether myocardial damage in these cases is a direct result of viral infection of cardiomyocytes, or if the association between the virus and myocarditis is temporal, while cardiac injury itself is caused by the maladaptive immune response to the ongoing infection or other mechanisms. This is especially puzzling in light of reports of EMB-confirmed myocarditis with visible necrosis in the course of SARS-CoV-2 infection, but no viral RNA present in the samples [82]. Experimental studies, however, have shown that SARS-CoV-2 can induce cytopathic effects such as contractility loss and even cell death in tissue models [81,83]. Hence, the mechanisms responsible for myocardial injury in COVID-19-associated myocarditis are unclear and require further research. Endothelial cell infection and subsequent endotheliitis could be yet another mechanism responsible for cardiac injury in the course of COVID-19 that should also be considered [35].

### 3.3. Immune Response to the Viral Infection

Immune response to the viral infection and its mechanisms implicated in the persistence and severity of myocarditis remain only partially described. The most common immune infiltrate observed in viral myocarditis is lymphocytic, and it almost always also includes macrophages [84]. At the beginning, immune response involves innate mechanisms, since natural killer cells followed by macrophages are usually the first to be recruited to the injured myocardium. Lymphocytes arrive later, and their infiltration is most pronounced at 7–14 days, which corresponds with the most severe phase of the disease. Using CVB3-infected models, Opavsky et al. have demonstrated that CD4^−/−^ CD8^−/−^ mice, as well as TCRβ^−/−^ mice, had much better survival than control mice, confirming the notion of the importance of T cell response in host susceptibility [85]. Moreover, the CD8^−/−^ genotype increased disease severity, which by comparison was attenuated in CD4^−/−^. Interestingly, in the same experiment, regardless of the mouse genotype, virus titers were similar and were not associated with improved outcomes. Work by Shi et al., on the other hand, highlights the importance of regulatory T (T_reg_) cells, which were shown to protect mice from CVB3-induced myocarditis [86]. These findings show the complex interplay between T cell subpopulations and demonstrate the detrimental role of T cell response in the exacerbation of the disease. In vitro and in vivo experiments suggest that p56^lck^ kinase expression appears to be a crucial host factor for CVB3-mediated heart disease [44]. There are substantially fewer data regarding the role of B cells, but they also are thought to possibly contribute to myocarditis thorough multiple mechanisms such anti-heart antibody production and the mobilization of other immune cells [4]. While viruses may be cleared from the myocardium, immune-mediated myocardial injury is suspected to play a major role in the progression of myocarditis [32]. Response against autoantigens as well as molecular mimicry between infectious agents’ antigens and cardiac tissue antigens are thought to play a role in the development of the disease [87]. There are multiple lines of evidence from mice and human studies that suggest the possible significance of these phenomena, and they have been extensively discussed by other authors [18,87,88,89,90]. As the line between the role of viruses and autoimmunity in virus-positive myocarditis is not clear, some of the mechanisms that may also play role in patients with positive viral PCR in EMB are described in further sections, which relate to autoimmune myocarditis. It is also important to note that in vitro, cardiac fibroblasts have been shown to play a role in the secretion of pro-inflammatory cytokines and maintenance of viral load, indicating that cardiac cell types other than cardiomyocytes may also contribute to the immune response and disease severity [91,92].

### 3.4. Clinical Implications of Viral Etiology

Clinical implications of viral presence in the myocardium remain not entirely understood, since conflicting reports exist regarding the prognostic role of the presence of viral genomes in EMB samples. Moreover, their applicability in the context of myocarditis is limited since some reports included patients with diagnoses different from myocarditis, such as DCM [33,93,94,95]. The best available evidence so far comes from a study by Kindermann et al., which suggests that the presence of viruses is not related to poor patient outcomes [33]. More recently, the transcriptional activity of viruses has been suggested to be of much importance; but, data regarding the validity of this hypothesis right now are scarce [36,63,64,65]. It is, however, important to highlight that viruses may be cleared from the myocardium (usually after the acute phase of the disease), which influences therapeutic approaches available to patients, as treatments for autoimmune myocarditis become an option [32].

Antiviral therapy in viral myocarditis remains controversial, and so far, interferon-β (IFN-β) has received the most attention as a potential candidate for such use. However, IFN-β use for viral myocarditis treatment is not endorsed by international guidelines, as data supporting such therapy are very limited and based on cohorts without active myocarditis, thereby making its applicability doubtful in the acute setting [2,5,8,96,97,98]. The routine use of intravenous immunoglobulin (IVIG) is also not supported by robust evidence, as the results of available meta-analyses are conflicting, and the patients included in the studies were not always diagnosed using EMB or CMR [99,100,101]. Therefore, it is not routinely recommended by international societies [2,5,8]. To date, international guidelines state that specific antiviral agents (such as acyclovir, ganciclovir) may be used for herpes virus myocarditis, even if “off-label”; however, robust evidence for their use is lacking, and it is strongly recommend to consult an infectious disease specialist if the use of antiviral agents is considered [2,5,22,102].

According to the 2013 Position Statement of the ESC, a positive viral PCR on EMB is a major contraindication to immunosuppression [2]. The 2021 HF guidelines of the ESC highlight that immunosuppressive therapy should not be routinely used in acute myocarditis without evidence of autoimmune disease [22]. However, in cases of high suspicion of immune-mediated myocarditis, empirical administration of intravenous corticosteroids may be taken into consideration before the results of EMB become available, especially in presence of complications such as acute HF, malignant arrhythmias and/or high degree atrioventricular (AV) block (i.e., fulminant myocarditis). The 2020 Expert Consensus Document of the American Heart Association (AHA) states that corticosteroids can be considered in fulminant myocarditis or complicated acute myocarditis, and the therapy may be maintained in patients with B19V/HHV-6 infections of the myocardium in case of a good response to treatment or low viral load [5,103].

HF in the context of myocarditis should be treated with standard medical therapy [2,5]. Similarly, arrhythmia treatment principles do not differ significantly from other patient populations; therefore, rhythm abnormalities should be managed according to the available guidelines [2,5,22,104,105,106]. Mechanical circulatory support should be considered in patients with particularly fulminant presentations.

## 4. Myocarditis in the Course of Parasitic Infections

### 4.1. Parasitic Involvement in Cardiac Disease

Myocarditis can be caused by a wide range of protozoa and helminths, with *Trypanosoma* spp. appearing as the most relevant etiological agents [107]. Trypanosomiasis in cardiovascular disease is mostly associated with Chagas’ disease (also referred to as ‘American trypanosomiasis’), which is a zoonosis caused by *Trypanosoma cruzi,* a protozoan, obligate intracellular parasite. It is a neglected tropical disease that is endemic in all Latin American countries; however, with increasing population mobility, cases in non-endemic regions are also being reported [108]. As it stands, it is estimated that 6–7 million people are infected with *T. cruzi* worldwide, and up to 30% of them will develop Chagas’ cardiomyopathy which is associated with HF, arrhythmia, stroke, thromboembolism and sudden death [108,109]. Interestingly, human African trypanosomiasis (HAT; sleeping sickness), caused by *Trypanosoma brucei gambiense* and *Trypanosoma brucei rhodesiense*, is associated with myocarditis as well [110,111]. Other parasitic infections such as trichinellosis, toxoplasmosis or cysticercosis can also lead to development of myocarditis [107]. Here, American trypanosomiasis receives the most attention because of its high prevalence in global population and significant frequency of cardiac complications.

### 4.2. Pathogenesis and Clinical Picture of Chagas’ Disease

The pathogenesis of Chagas’ disease has been extensively studied in animal models and clinical observation. Since myocarditis constitutes only one particular aspect of the systemic syndrome and not the totality of Chagas’ disease, it is important to consider cardiac involvement in the context of the entire natural course of the disease. *T. cruzi* infections are usually divided into two consecutive stages referred to as acute and chronic Chagas’ disease [112]. The acute phase, is associated with parasitemia observable in direct examination of the blood and subsequent chronic phase, is characterized at first by lack of symptoms and, later, by severe gastrointestinal and cardiac manifestations [108]. *T. cruzi* is transmitted by Triatomine vector species (‘kissing bugs’), but importantly, it can also spread through blood transfusions or congenital infection. At first, trypomastigotes replicate near the inoculation site and subsequently spread throughout the body [113]. Trypomastigotes enter the cell where they differentiate into amastigotes and divide through binary fission [114]. Thereafter, amastigotes differentiate into trypomastigotes and disrupt the cell to infect surrounding cells. Alternatively, amastigotes can disrupt a cell prematurely, thus leading to their release, and reinvade cells through phagocytosis. While the parasite can infect any nucleated cell, it exhibits tropism towards cardiac and skeletal muscle cells. Experimental data suggest that such tropism may be related to well-developed plasma membrane repair mechanisms which facilitate cell entry of trypomastigotes [114]. *T. cruzi* infection of myocardial fibers results in visible cell damage and is associated with mononuclear cell infiltration. These infiltrates consist primarily of T cells and macrophages but can also include other immune cell types such as lymphocytes, eosinophils, neutrophils or plasma cells [113]. During the acute phase, the disease is often characterized by mild symptoms (i.e., splenomegaly, fever, malaise) [108,112]. However, a small subset of patients with fulminant acute disease presents with myocarditis, pericardial effusion and meningoencephalitis. If the acute phase is detected, it is frequently associated with various ECG changes. The acute phase usually resolves spontaneously after 4–12 weeks and is followed by the indeterminate phase of chronic Chagas’ disease characterized by detectable anti-*T. cruzi* antibodies and a lack of symptoms. This makes the disease particularly difficult to diagnose, considering that non-invasive diagnostic tools (i.e., ECG, radiograms, echocardiograms) are frequently normal [107,108,110,112]. It is important to note that a lack of symptomatic manifestation does not rule out subclinical myocarditis, as *T. cruzi* is found in the myocardium of patients with chronic disease [115]. Moreover, noninvasive testing such as echocardiography or CMR may in fact be abnormal, revealing ventricular dysfunction and/or myocardial fibrosis [116,117,118]. About 30–40% of patients will develop a determinate form of chronic disease, usually 10–30 years post-infection [119]. This phase of the disease is associated with gastrointestinal (e.g., megaesophagus, megacolon) and cardiac manifestations. Cardiac involvement leads to Chagas’ cardiomyopathy which hemodynamically resembles DCM but can be distinguished by a characteristic fibrosis pattern distribution (predominantly in apical and posterior regions of the left ventricle [LV]) and the involvement of the conduction system [112]. Moreover, Chagas’ heart disease is usually accompanied by chronic myocarditis, which concurrently with other factors (e.g., microcirculation pathologies, neurogenic dysfunction) contributes to progression of cardiac dysfunction [112,113,120,121]. In the chronic phase of the disease, myocarditis is also thought to be driven primarily by maladaptive immune response targeted at the parasite and host myocardium [121]. Similarly to previously described etiologies, autoimmunity against host antigens or molecular mimicry may play a significant role [122]. The importance of presence of *T. cruzi* in cardiac tissue during chronic Chagas’ disease remains not entirely understood. Available data suggest the presence of parasite DNA in hearts of both symptomatic (Chagas’ cardiomyopathy) and asymptomatic (indeterminate) patients [123]. What is more, the presence of *T. cruzi* has been shown to not clearly correlate with inflammation of myocardium in explanted and autopsy hearts [124]. Progressive cardiac tissue dilation, rhythm abnormalities, ventricular aneurysms and conduction system pathologies associated with Chagas’ cardiomyopathy lead to serious clinical manifestations such as HF, arrhythmias or thromboembolisms [112,125].

### 4.3. Management of Patients with Chagas’ Cardiomyopathy

The current evidence suggests that the prognosis of patients with Chagas’ cardiomyopathy remains worse than in the case of other etiologies [126,127,128]. Antitrypanosomal medication, i.e., benznidazole (first-line treatment) and nifurtimox, is recommended for all patients with acute Chagas’ disease [112]. The benefits of such therapy are less clear in the indeterminate stage of the disease, and the current Scientific Statement of the AHA does not recommend the routine use of antitrypanosomal agents in these patients, but rather it highlights that it could be offered. Benznidazole also did not show any significant effect on clinical outcomes in patients with established Chagas’ cardiomyopathy [129]. Therefore, AHA recommends that the management of patients with Chagas’ cardiomyopathy should be based on treatment regimens analogous to those observed in other HF etiologies; however, their efficacy in this condition is unknown [22,106]. International guidelines indicate that ventricular arrhythmias in American Trypanosomiasis can be treated with anti-arrhythmic drugs (i.e., amiodarone and beta-blockers) and electrophysiological interventions (i.e., catheter ablation, implantable cardioverter-defibrillator implantation) [104,105,112].

### 4.4. Cardiac Involvement in Other Parasitic Diseases

While Chagas’ disease remains notorious for cardiac involvement, in HAT, neurological problems dominate the clinical picture [110]. *T. brucei gambiense* and *T. brucei rhodesiense* are transmitted by tsetse flies (*Glossina* spp.), but similarly to *T. cruzi*, congenital and transfusional modes of transmission are also possible [107]. Cardiac disease in HAT has been studied much less extensively than in Chagas’ disease; however, studies have shown evidence of myocarditis in patients infected with *T.b. gambiense* and *T.b. rhodesiense* [130,131]. Moreover, Blum and colleagues noticed ECG abnormalities in 71% of HAT patients [132]. The same study, however, indicated that major conduction abnormalities are rare and frequently resolve after treatment. Interestingly, the pathogenesis of cardiac abnormalities in HAT appears to be different from Chagas’ disease, as parasites in HAT do not enter myocardial cells but are found interstitially [110]. Reliable long-term data on cardiac involvement in HAT are very scarce, with available reports only suggesting minimal long-term cardiac involvement in patients [133]. Moreover, it should be highlighted that myocarditis in the course of any parasitic infection does not have to be caused by the presence of the pathogen in the myocardium, but rather by a powerful inflammatory response to the parasite (e.g., helminths) resulting in reactive eosinophilia and subsequent eosinophilic myocarditis, which is associated with poor prognosis [134,135]. The exact mechanisms involved in this process are still unknown.

## 5. Bacterial Myocarditis

### Overview

Bacteria are considered to be an uncommon cause of myocarditis and even if there are case reports of such infections, a recent study has shown that many of these reports do not rely on autopsy/EMB histology, and a large portion of them do not even include CMR [136]. Therefore, their credibility is questionable, and no clear clinical guidelines regarding such cases are available due to the scarcity of data. Nevertheless, it is important to consider the possibility of myocarditis triggered by bacterial species (e.g., *Borrelia* spp., *Corynebacterium* spp., *Streptococcus* spp., *Staphylococcus* spp.) [136,137]. Carditis in the course of Lyme disease appears to be more common, as its estimated incidence in patients with Lyme disease is 0.3–4% [138]. Myocarditis during diphtheria is also assumed to be more common in countries without widespread immunization [137]. Therefore, they can be a significant cause of myocarditis in certain populations. In such cases, appropriate antimicrobial therapy should be introduced in order to target the underlying cause of the disease [22].

## 6. Autoimmune Myocarditis and Drug-Induced Myocarditis

### 6.1. Overview

Autoimmune myocarditis is diagnosed in the presence of immunohistological evidence of myocarditis with negative viral PCR, with or without serum anti-heart antibodies [2]. These patients present yet another heterogenous group, because autoimmune myocarditis can occur with exclusive cardiac involvement as well as in the course of an immune-mediated disease such as sarcoidosis, systemic sclerosis (SSc), systemic lupus erythematosus (SLE) or eosinophilic granulomatosis with polyangiitis (EGPA) [139]. It can also be induced by exposure to various substances (e.g., drugs, alcohol, vaccines) [4]. Moreover, as has been previously described, virus-induced myocarditis appears to possess a significant autoimmune component, which may not resolve after viral clearance and present as autoimmune myocarditis.

### 6.2. Pathogenesis and Genetic Predisposition to Autoimmune Myocarditis

In many patients with autoimmune myocarditis, the exact trigger and molecular interactions underlying the disease are unknown [140]. However, it appears that a lack of balance in T cell populations may be responsible for this owing to a disturbed balance of pro- and anti-inflammatory activity of the immune system. This hypothesis is consistent with mice models that exhibit susceptibility to the development of experimental autoimmune myocarditis. In a study by Chen et al., a more susceptible mouse strain was shown to have higher percentage of CD4^+^ T cells along with a tendency to differentiate into the Th17 phenotype and a lower frequency of T_reg_ cells when compared to a less susceptible strain [141]. Interestingly, in experimental models of autoimmune myocarditis, the Th17 response has been implied to play a critical role in autoimmune driven progression of myocarditis to DCM [142,143]. Genetic features are also thought to play a role in predisposing to autoreactivity. These include both genes related to the major histocompatibility complex (MHC) as well as genes independent of the MHC [140]. Several of these associations between genetic variants and myocarditis have been described [140,142]. Genetic predisposition may not only determine susceptibility but also the course of the disease. Recently, patients with clinically suspected myocarditis and desmosomal and sarcomeric gene variants have been shown to have a higher frequency of adverse cardiovascular events when compared with patients without such variants [144,145]. Further studies will be needed to investigate biopsy-proven myocarditis patients, with various clinical presentation, to determine if genetic variants associated with cardiomyopathies may influence prognosis apart from the classical predictors [146,147].

Similarly to virus-positive myocarditis, molecular mimicry as well as exposure and subsequent immune response against intracellular antigens are also hypothesized to play a role. Anti-heart antibodies may be present and are more frequently associated with acute phases of the disease [10]. Cardiac autoantibodies against autoantigens such as cardiac myosin, troponin or beta1-adregenic or M2 muscarinic receptors among others are suggested to play a direct pathogenic role in the immune-mediated phase of myocarditis [87]. However, it is important to note that the exact mechanisms that underlie autoimmune myocarditis may vary, particularly in the course of different immune-mediated diseases (e.g., immune-complex mediated in SLE) [139]. More recently, microbiota-derived peptide mimicry has been demonstrated to drive inflammatory cardiomyopathy in mice models of spontaneous autoimmune myocarditis, revealing yet another area for future research, as antibiotic treatment dampened immune response and prevented lethal cardiomyopathy [148]. What is important to note is that autoimmune myocarditis appears much more heterogenous when it comes to the characteristics of cellular infiltrates present in the myocardium when compared to viral myocarditis. Giant cell myocarditis (GCM), which is associated with the highest risk of death and heart transplantation, is considered to be the prototype of autoimmune myocarditis and has in fact been defined as “the most fatal of autoimmune diseases” [7,11,149]. On histology, GCM is characterized by infiltrates consisting of T cells and multinucleated giant cells typically associated with either intact or degranulated eosinophils [142,150]. Eosinophilic myocarditis, which is also characterized by poor prognosis when compared to the lymphocytic form, can have an autoimmune background as well [135,142]. Apart from infections, especially parasitic ones, eosinophilic myocarditis has been reported to be associated with immune-mediated diseases such as EGPA [135]. EMB samples from patients with myocarditis caused by various immune cell infiltrates are presented on Figure 2.

### 6.3. Pathogenesis of Drug-Associated Myocarditis

Involvement of drugs in myocarditis is complex, and definitions are not clear. The 2020 AHA Expert Consensus Document defines drug-induced myocarditis as caused by direct cytotoxic effect of the drug [5]. This, however, is not the only mode of drug involvement in myocarditis, as hypersensitivity reactions and more complex mechanisms such as those relating to immune checkpoint inhibitor (ICI)-associated myocarditis or vaccine-induced myocarditis are also to be considered.

Hypersensitivity reactions to drugs and drug reaction with eosinophilia and systemic symptoms (DRESS) are a possible cause of eosinophilic myocarditis [5]. They can be caused by exposure to medications such as clozapine, carbamazepine, minocycline, β-lactam antibiotics and even vaccination. Similarly to descriptions regarding other etiologies, in this case, eosinophilic myocarditis is also most likely caused by a maladaptive immune response resulting in eosinophilia and subsequent myocardial damage [134].

Special attention should be given to ICI-associated myocarditis as it is considered to be the most frequent immune-related adverse event during ICI treatment and carries high mortality [151]. ICIs are a group of monoclonal antibodies which enhance the host immune response against cancer cells by inhibiting key immunoregulatory mechanisms (i.e., checkpoints) [152]. Such drugs can target cytotoxic T lymphocyte-associated antigen 4 (CTLA-4), programmed death-1 receptor (PD-1) and its ligand (PD-L1) and lymphocyte-activation gene 3 (LAG-3). While the pathophysiology of ICI-associated myocarditis is still poorly understood, some work has been conducted to identify the putative mechanisms behind this etiology. EMB findings appear to be consistent and show lymphocytic infiltrates, while the main risk factor appears to be combination ICI therapy (e.g., anti-CTLA-4 and anti-PD-1) [151,153,154]. Current evidence from mice and humans suggests that myocarditis resulting from ICI use is driven by CD8^+^ T cell reactivity against autoantigens such as α-myosin [155]. Experimental data and clinical observations also suggest that female sex could be a predisposing factor for ICI-myocarditis through the inhibition of estradiol-dependent expression of MANF (Mesencephalic Astrocyte Derived Neurotrophic Factor) and HSPA5 (Heat Shock 70 kDa Protein 5) in the heart during treatment [156,157].

Vaccine-associated myocarditis has gathered a lot of attention because of its reports in association with COVID-19 vaccines. As described previously, hypersensitivity is one of the possible pathophysiological mechanisms behind adverse cardiovascular effects of vaccines, and there are reports of histologically documented cases of eosinophilic myocarditis following mRNA vaccination against COVID-19 [158]. The characteristics of infiltrates are, however, heterogenous, and histological investigation has so far shown many possible presentations of myocarditis following COVID-19 vaccination, such as GCM or lymphocytic myocarditis [159,160]. It has been demonstrated that myocarditis is more common in younger men and after sequential doses of COVID-19 vaccine, while its risk is highest after a second dose of the mRNA-1273 vaccine [161]. The mechanisms responsible for myocarditis following mRNA vaccination against COVID-19 are unknown, but current hypotheses imply maladaptive immune response possibly modified by immune-genetic background, age, sex and hormonal differences [162]. Circulating spike protein was found to be present in patients with post-COVID-19 mRNA vaccine myocarditis, giving a possible hint into the underlying cause of this adverse reaction [163]. Available evidence does not suggest that molecular mimicry plays a significant role [164]. Aberrant cytokine-driven lymphocyte cytotoxicity and profibrotic myeloid cell response appear to be key in the immunopathogenesis of vaccine-associated myocarditis [165].

### 6.4. Clinical Implications of Autoimmune and Drug-Associated Myocarditis

GCM and eosinophilic myocarditis have the worst prognosis, but it is important to remember that lymphocytic myocarditis also can have a fulminant presentation [7,8,135,142,150]. In case of virus-negative myocarditis or complicated clinical presentation and high-suspicion of immune-mediated disease, immunosuppressive treatment can be considered in addition to standard guideline-directed medical therapy for HF [2,5,22]. However, routine use of corticosteroids before obtaining a definitive diagnosis is not endorsed by either ESC or AHA. Some randomized trials reported promising improvements in LVEF with the best response in virus-negative patients [166,167,168,169,170].

Immunosuppressive therapy should be modulated on the histological type of myocarditis. Currently, for lymphocytic virus-negative myocarditis, the use of prednisone and azathioprine is supported by the best available evidence. As such, corticosteroid therapy combined with the use corticosteroid-sparing agent is the most used first-line therapy regimen and should be tailored depending on patient characteristics (e.g., underlying autoimmune disease) [139]. There are ongoing clinical trials that may provide randomized multicenter evidence for the use of immunosuppressive treatment in the future [171]. Immunosuppressive therapy in myocarditis should be considered not only after confirmation of the absence of a viral genome on EMB, but also after an exhaustive search for potential contraindications to the treatment itself. In fact, a detailed “safety checklist” has been published with the purpose of identifying absolute and/or relative contraindications to immunosuppressive therapy and defining the individual risk of each patient [172]. Apart from immunosuppression, similarly to virus-positive myocarditis, HF and arrythmias should be managed according to the available guidelines [2,5,22,104,105,106]. The utility of anti-heart antibody detection in patients remains unclear [10].

Giant cell and eosinophilic myocarditis are rare, but rapidly progressing myocarditis histotypes may, and may in turn, provide the most dramatic improvement of prognosis if the diagnosis is made early and aggressive immunosuppression is started. The treatment of GCM, which has a dramatic clinical presentation that includes progressive HF, cardiogenic shock and ventricular arrythmias, is based on the prompt initiation of combination immunosuppressive therapy [150,173,174]. It should involve intravenous corticosteroids and at least one, and most often two, other immunosuppressive agents, such as azathioprine and cyclosporine [150]. The decision on how to stop therapy, in case of remission, is still not defined and needs to be individualized for each case [175]. As there is a risk of recurrence in patients with GCM, low-dose calcineurin inhibitors are sometimes continued indefinitely, although there is no strong evidence supporting such practice [150]. Due to its rarity, the management of GCM is largely based on registry data.

Regarding eosinophilic myocarditis, its therapy is also strongly based on corticosteroids; in accordance with the response, their dosage may be carefully tapered during follow up, and a steroid-sparing agent may be added, such as azathioprine or mycophenolate mofetil [5,176]. Importantly, if eosinophilic myocarditis appears in the context of EGPA, as its first presentation or in already-diagnosed patients, medical therapy should be modulated taking into account the underlying systemic disease [139]. It is important, however, that the approach to myocarditis patients is always personalized, as in some cases, clinicians’ attention should be directed to primarily resolve the underlying cause of the disease (e.g., stop exposure to substances causing hypersensitivity reactions) [5,22].

Interestingly, ICI-associated myocarditis presents a distinct phenotype which is often characterized by preserved LVEF (in over 50% of patients), HF as well as severe ventricular and supraventricular arrhythmias [151,177,178,179,180]. While the immediate withdrawal of ICI treatment and high-dose intravenous corticosteroid therapy is recommended for these patients, the mortality remains high, with 50% of patients dying according to published retrospective studies [5,181,182]. There are reports of other therapies such as anti-thymocyte globulin (anti-CD3 antibody), abatacept (CTLA-4 agonist), alemtuzumab (anti-CD52 antibody) and tofacitinib showing promising results; however, they have not been tested in large trials and require further investigation [182,183,184,185].

Myocarditis following vaccination against COVID-19 has limited data on outcomes. Available data from a study by Kracalik et al. suggest that myocarditis relating to mRNA vaccination among adolescents and young adults has a high rate of recovery at least 90 days since onset of myocarditis (81% designated as fully recovered); however, out of 393 examined patients, 65 have not fully recovered [186]. Moreover, even in the group designated as recovered by their healthcare provider, 48% of surveyed patients reported still experiencing various symptoms (64% in the not recovered group) such as chest pain, fatigue, shortness of breath and palpitations (median interval from myocarditis onset to survey completion was 143 days). Importantly, a quarter of all patients examined were prescribed daily medications (e.g., colchicine, beta-blocker, non-steroidal anti-inflammatory drugs, angiotensin-converting enzyme inhibitors, diuretics, corticosteroids, angiotensin II receptor blockers). In an analysis by Ammirati et al., among 75 patients with an available follow-up, no patient died or required hospitalization after a median time of 147 days [187]. A total of 10.6% of patients with an available follow-up CMR (n = 49) had a dilated LV, while 20.4% had persistent edema, and 79.6% had a residual scar based on late gadolinium enhancement. There is a need for further investigation when it comes to pathogenesis, management and long-term outcomes of such patients.

Particularly in the course of immune-mediated disease, autoimmune myocarditis can vary greatly in presentation, and first signs of cardiac involvement are not easy to outline. In SSc, for instance, left ventricular systolic dysfunction is considered to be the hallmark of cardiac involvement, which makes routine echocardiography particularly useful [139]. On the other hand, cardiac sarcoidosis may present with arrhythmic manifestations (AV block, ventricular tachycardia) in otherwise asymptomatic subjects [139,188,189]. Therefore, cardiac involvement in systemic immune-mediated disease is heterogenous, and therapy as well as patient management should be tailored to the individual patient in concordance with the best available evidence and guidelines.

## 7. Conclusions

Diagnosis, management and prognostic stratification of myocarditis patients remain difficult. Even with progress being made in experimental and clinical research, underlying mechanisms of the disease are still only partially understood. Current evidence suggests that myocarditis is not only heterogenous in its clinical presentation but also in processes governing its pathogenesis depending on etiology. A thorough understanding of these issues is needed in order to optimize patient care, implement new therapeutic strategies as well as to better understand factors that influence clinical presentation, patient outcomes and response to treatment. This requires the development of new animal models and further experimental research that will examine the role of host (e.g., genetic, immunity-related) and environmental (e.g., infectious) factors in myocarditis. Moreover, clinical evidence that regards risk stratification depending on clinical features (e.g., positive viral PCR in EMB) is required to better understand the disease course and improve patient care. Finally, more evidence from multicenter randomized controlled trials is required when it comes to currently applied or proposed and management strategies.

## Figures and Tables

**Figure 1 biology-12-00874-f001:**
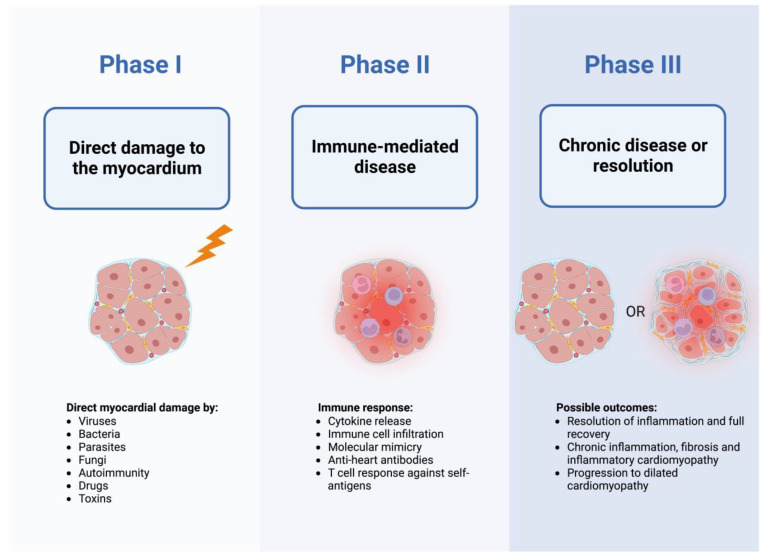
Triphasic model of myocarditis pathogenetic mechanisms based on clinical observations and experimental data [9,13,14,15,16,17,18,19]. Created with BioRender.com.

**Figure 2 biology-12-00874-f002:**
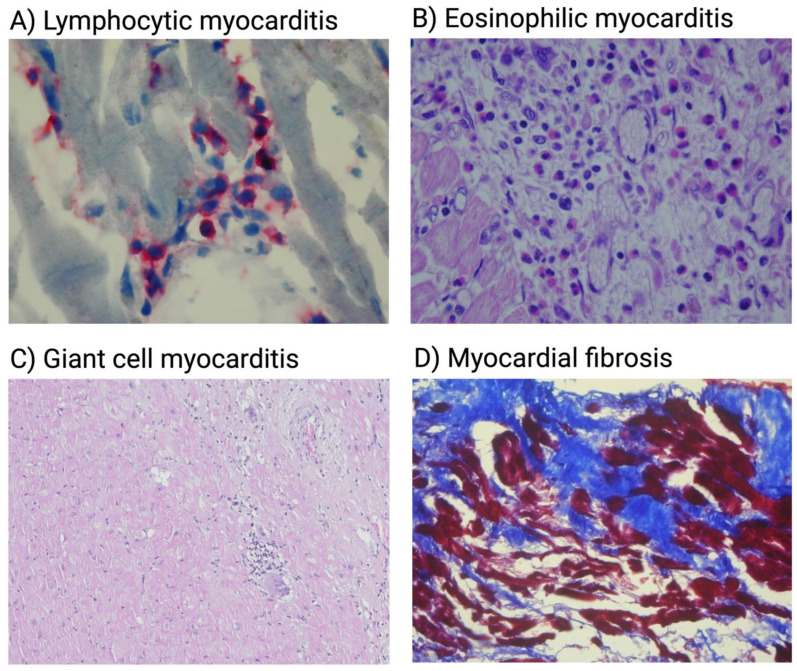
Histological classification of myocarditis on EMB. (**A**) Acute lymphocytic myocarditis. An infiltrate rich in CD3(+) lymphocytes (red stain) with concomitant myocytolysis is visible. (**B**) Eosinophilic infiltrate present in biopsy specimen of patient with eosinophilic myocarditis. (**C**) Giant cell myocarditis with visible area of inflammation and giant cell infiltration. (**D**) Massive myocardial fibrosis (blue) visualized using Masson’s trichrome stain. Created with BioRender.com.

**Table 1 biology-12-00874-t001:** Viruses frequently associated with myocarditis [4,28,33].

Virus	Tropism or Tissue Toxicity	Genome
Parvovirus B19	Vasculotropic	ssDNA
Enteroviruses	Cardiotropic	(+) ssRNA
Adenoviruses		dsDNA
Human herpesvirus type 6	Lymphotropic	dsDNA
Epstein–Barr Virus		
Cytomegalovirus		
Hepatitis C virus	Cardiotoxic	(+) ssRNA
Human immunodeficiency virus		
Influenza viruses		
SARS-CoV-2	ACE2-tropic	(+) ssRNA

ACE2, angiotensin-converting enzyme 2; dsDNA, double-stranded DNA; SARS-CoV-2, severe acute respiratory syndrome coronavirus-2; ssDNA, single-stranded DNA; (+) ssRNA, positive-sense single-stranded RNA.

## Data Availability

Publicly available datasets were analyzed in this study.

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
