# Peer review of "Myocarditis: Etiology, Pathogenesis, and Their Implications in Clinical Practice"

_biology, 2023, doi:10.3390/biology12060874_

Round 1

Reviewer 1 Report

Dear Authors,

thank you for submitting this review paper to Biology. I have read it with interest and I have the following comments:

- It is a well written overview of about myocarditis.

- As stated in the title, the etiology and pathogenesis have implications with clinical presentation and therapy; from a clinical point of view it would be of value include for each type a paragraph about therapy and controversies

- as myocarditis is a complex diagnosis, sometimes of exclusion, I'd sense at the beginning of the text to present clinical diagnostic suspition features and the role of EMB and alternative or emerging diagnostic tools.

- at the end of the text a summarising paragraph of future direction in terms of diagnosis, emerging pathology and controversies will add value to the paper

The Text is well written and sounding

Reviewer 2 Report

The review “Myocarditis: etiology, pathogenesis and their implications in clinical practice” by Brociek et al. discusses the possible etiologies of myocarditis, outlines the key processes governing its pathogenesis and summarizes best available evidence regarding patient outcomes and state-of-the-art therapeutic approaches of the disease.

This is a well-written manuscript. The topic of the study is of high clinically relevance.

However, I have the following major remarks on the manuscript:

A crucial point is the insufficiently described data on viral infection. This concerns for example the most common erythroparvovirus. The authors do not address the presumably irrelevant latent infection in contrast to transcriptionally active virus and progenosis. This point could be crucial with regard to possible later therapy initiation. Here, a differentiated consideration of the differential therapy is missing. Clinical antiviral studies (e.g. interferon-b) should be cited.

Important in this point is a critical consideration of non-invasive diagnostics, which makes virus detection not possible at all. 

More citations should be made for the SarsCoV-2 genomes detection part.

Reviewer 3 Report

I read with great interest this manuscript reviewing the pathophysiology and treatment of myocarditis.

Major points:

1. I would like to see more emphasis on the few treatable/modifiable viral infections. Giant cell myocarditis caused by parvovirus B19 infections has an unfavorable clinical outcome and should be treated even in the absence of clear data guiding this approach.

2. A major determinant of poor left ventricular function in the chronic phase is genetic predisposition. In particular variants and mutations in cardiomyopathy-related genes may need a "second hit" such as viral infections to manifest the disease. Please add a paragraph on this issue.

3. Immune checkpoint inhibitor-associated myocarditis is evolving as the most common form of myocarditis. I suggest to expand the description on this topic. It might be wise to put it in a distinct chapter to emphasize its importance. It might also be good for readability to add a table with the known agents and the known clinical sequelae. The paragraph on treatment should be expanded.

4. I would like to see a paragraph on new or evolving therapies such as beta-adrenergic receptor antibody scavengers.

Minor point:

1. I would suggest to add COVID-19 to table 1.

In general, the manuscript is well-written, a last round of editing will be sufficient.

Round 2

Reviewer 3 Report

I am satisfied with the changes made to the manuscript